# The Clinical Significance of the Hyperintense Acute Reperfusion Marker Sign in Subacute Infarction Patients

**DOI:** 10.3390/diagnostics11112161

**Published:** 2021-11-22

**Authors:** Ji Young Lee, Kyung Mi Lee, Hyug-Gi Kim, Ho-Geol Woo, Jin San Lee, Eui Jong Kim

**Affiliations:** 1Department of Medicine, Graduate School, Kyung Hee University, #23 Kyungheedae-ro, Dongdaemun-gu, Seoul 02447, Korea; mail2jiyoung@gmail.com; 2Department of Radiology, College of Medicine, Kyung Hee University Hospital, Kyung Hee University, #23 Kyungheedae-ro, Dongdaemun-gu, Seoul 02447, Korea; khyukgi@gmail.com (H.-G.K.); euijkim@hanmail.net (E.J.K.); 3Department of Neurology, College of Medicine, Kyung Hee University Hospital, Kyung Hee University, #23 Kyungheedae-ro, Dongdaemun-gu, Seoul 02447, Korea; nr85plasma@naver.com (H.-G.W.); xpist@naver.com (J.S.L.)

**Keywords:** subacute infarction, postcontrast FLAIR, HARM sign, ischemia

## Abstract

Purpose: The hyperintense acute reperfusion marker (HARM) is characterized by the delayed enhancement of the subarachnoid or subpial space observed on postcontrast fluid-attenuated inversion recovery (FLAIR) images, and is considered a cerebral reperfusion marker for various brain disorders, including infarction. In this study, we evaluated the cerebral distribution patterns of HARM for discriminating between an enhancing subacute infarction and an enhancing mass located in the cortex and subcortical white matter. Materials and methods: We analyzed consecutive patients who experienced a subacute ischemic stroke, were hospitalized, and underwent conventional brain magnetic resonance imaging including postcontrast FLAIR within 14 days from symptom onset, as well as those who had lesions corresponding to a clinical sign detected by diffusion-weighted imaging and postcontrast T1-weighted imaging between May 2019 and May 2021. A total of 199 patients were included in the study. Of them, 94 were finally included in the subacute infarction group. During the same period, 76 enhancing masses located in the cortex or subcortical white matter, which were subcategorized as metastasis, malignant glioma, and lymphoma, were analyzed. We analyzed the overall incidence of HARM in subacute ischemic stroke cases, and compared the enhancement patterns between cortical infarctions and cortical masses. Results: Among 94 patients with subacute stroke, 78 patients (83%) presented HARM, and among 76 patients with subcortical masses, 48 patients (63%) presented peripheral rim enhancement. Of 170 subcortical enhancing lesions, 88 (51.8%) showed HARM, and 78 (88.6%) were determined to be subacute infarction. Among 94 patients with subacute stroke, 48 patients (51%) had diffusion restrictions, and HARM was found in 39 patients (81.2%). Of the 46 patients (49%) without diffusion restriction, 39 patients (84.8%) showed HARM. Conclusions: The presence of HARM was significantly associated with subacute infarctions. For the masses, a peripheral rim enhancement pattern was observed around the mass rather than the cerebral sulci on postcontrast FLAIR.

## 1. Introduction

The evaluation of patients visiting the hospital with neurologic symptoms during the subacute period is important in determining the direction of treatment. Patients are usually examined based on the assumptions of physicians, which may include ischemic stroke, degenerative diseases, or tumorous conditions, and are taken through conventional imaging workups, such as computed tomography (CT) or magnetic resonance imaging (MRI). MRI is a sensitive modality, and T2-weighted imaging (T2WI) is more sensitive than T1-weighted imaging (T1WI) in detecting intracranial pathology. Diffusion-weighted imaging (DWI) and perfusion-weighted imaging (PWI) have increased the accuracy of the diagnosis of acute cerebral infarction, although subacute infarctions may not show diffusion restriction due to pseudonormalization. T1WI with gadolinium (Gd) administration can provide valuable information for diagnosing subacute infarctions [1,2]. Contrast enhancement in the central nervous system (CNS) is the result of a combination of the disruption of the blood-brain barrier (BBB), high vascularity, and contrast leakage into the lymphatic system [2,3,4,5]. After at least seven days, subacute infarcts show parenchymal enhancement due to the breakdown of the BBB [6].

Fluid attenuated inversion recovery (FLAIR) is also highly sensitive for the detection of ischemic lesions because it nulls the signal intensity from the cerebrospinal fluid as dark [7]. Intracranial pathologic conditions that present enhancement on T1WI usually show marked enhancement on postcontrast FLAIR [8]. The hyperintense acute reperfusion marker (HARM) is characterized as the delayed contrast enhancement of the subarachnoid or subpial space on postcontrast FLAIR images in various intracranial pathologic conditions, and is believed to be associated with permeability changes in the BBB. HARM is not a specific finding for acute stroke, and has been described in other BBB-disturbing conditions, such ascarotid artery stenting, post-operative state for cardiothoracic surgery, traumatic brain injury, and seizure [9,10,11,12,13]. HARM was found in 10–40% of patients with acute stroke in the previous studies [8,14], and is considered an useful marker for BBB disruption in acute stroke [15,16,17].

Despite the success of MRI in evaluating acute ischemia, the differential diagnosis of nodular enhancing lesions in the cortical and subcortical areas often remains challenging. In other words, the visible enhancement of subacute infarctions can be misdiagnosed for primary brain tumors or metastases. Smirniotopoulos et al. reported that if nodular subcortical lesions are detected, metastasis or septic emboli should be considered [18]. Small subcortical infarctions can sometimes share morphologic features with malignant lesions, and differentiating true ischemic lesions from malignant lesions may be difficult, even for experienced radiologists. Although DWI has been shown to be an excellent tool for the detection of acute stroke, its sensitivity would appear to have limitations, as witnessed by reports of false-negative cases. The prevalence rate of false negative DWI in patients with acute ischemic stroke is highly variable, ranging from 0% [19] to 25.6% [20,21,22,23,24,25,26,27,28,29], notwithstanding that the stroke lesions in these studies were not always confirmed radiologically. Furthermore, ischemic stroke mimics can occur on DWI with non-cerebrovascular neurological conditions [30], and this can make diagnosis even more challenging. PWI can also help prevent misinterpretation, however, this study is not routinely performed without clinical suspicion.

The discrimination between a true ischemic stroke and other conditions, including malignant primary CNS tumor or metastasis, is essential for planning adequate treatment, and estimating outcomes and future diagnosis. In this study, we evaluated the usefulness of HARM in differentiating mass-like enhancing subacute infarction from malignant lesions for cortical and subcortical enhancing nodular lesions on postcontrast T1WI. We hypothesized that a linear contrast enhancement pattern along the subpial and subarachnoid space adjacent to the enhancing nodular lesion on postcontrast FLAIR images suggests infarction rather than malignant lesions.

## 2. Materials and Methods

### 2.1. Patient Population

This retrospective, single-center study was approved by the institutional review board, and informed consent was obtained from all subjects involved in the study.

Strokes may be classified and dated as early hyperacute, late hyperacute, acute, subacute, or chronic. As a first step, we needed to agree on a common framework for defining what is meant by “acute” and “subacute” stroke. There is substantial heterogeneity in the terminology denoting time from the onset of stroke. The definitions of hyper-acute as 0–24 h, and acute as 1–7 days were affirmed by the international Stroke Recovery and Rehabilitation Roundtable, which link to the currently known biology of recovery [31,32]. Apparent diffusion coefficient (ADC) maps may depict darkening within minutes of stroke onset, and are more sensitive than diffusion weighted sequences. A good rule of thumb is that if the signal intensity on ADC maps is low, the stroke is less than 1 week old. Low signal intensity on ADC maps persists for about 7–10 days, and it can be observed in the early subacute stage. In most infarcts, parenchymal enhancement on postcontrast T1WI is seen between 1 week and 2 months after stroke [33]. In this study, acute stroke was defined based on these radiologic findings and clinical symptoms. We classified the patients with diffusion restriction (low signal intensity on ADC maps, and high signal intensity on DWI), with no parenchymal enhancement on postcontrast T1WI, and with neurologic symptoms within 7 days of onset as having acute stroke, and excluded those cases from the study.

Between May 2019 and May 2021, we retrospectively reviewed a total of 12,582 MRIs for searching the cortical enhancing lesions. Based on the picture archiving communication system (PACS) and electronic medical records (EMR) results, 275 patients who had enhancing lesions in the brain were identified and were subdivided into stroke (199 patients) and brain tumors (76 patients). Among 199 patients who had diagnosed infarction, a total of 69 patients were excluded for the following reasons: (1) 26 had been diagnosed with acute infarction; (2) 6 patients had an enhancing mass combined with infarction; (3) 1 patient had meningitis; (4) 1 patient had demyelinating disease; (5) 3 patients had a previous history of surgery; and (6) the brain lesion in 32 patients did not meet the available reference standard due to poor image quality with motion artifacts. Furthermore, 36 patients were additionally excluded because subacute infarctions were located in the periventricular white matter, deep gray matter, or infratentorium, including the brainstem and cerebellum. Subsequently, 94 patients with subacute cortical infarction were included. We characterized pathology based on the typical MRI findings, clinical courses, and stroke treatment outcomes for a minimum of 1 month. Follow-up MRIs were obtained more than once, if necessary, in challenging cases. The details of this cohort are shown in Table 1. During the same period, a total of 76 patients diagnosed with enhancing masses were also identified. The masses located in the cortex or subcortical white matter were analyzed. The enhancing masses were subcategorized as metastasis (*n* = 66), malignant glioma (*n* = 7), and lymphoma (*n* = 3).

Presumed subacute infarction patient evaluation was performed by an experienced stroke neurologist. Symptom severity was evaluated using the National Institutes of Health Stroke Scale (NIHSS) scores at admission and on discharge. Clinical outcomes were evaluated using the modified Rankin Scale (mRS) score on discharge. Other information, such as patient demographics, medical history, and risk factors, were obtained directly from the web-based stroke registry, known as the prospective stroke registry of the fifth section of the Clinical Research Center for Stroke (CRCS-5). The stroke onset time was defined as the last known time at which the patient was free of symptoms. Potential stroke mechanisms were determined using the conventional Trial of Org 10172 in Acute Stroke Treatment (TOAST) classification.

### 2.2. MRI Acquisition

MRI was performed on a 3 Tesla (T) MRI system (VIDA, Siemens or Achieva TX; Philips Medical Systems, Best, The Netherlands) equipped with a 64- or 32-channel head coil. The protocol for conventional brain MRI evaluation included DWI, ADC maps, three-dimensional magnetization-prepared rapid gradient echo T1WI (3D MPRAGE T1WI), T2WI, FLAIR, susceptibility-weighted imaging (SWI), 3D postcontrast T1WI, postcontrast FLAIR, and vascular imaging, including time-of-flight (TOF) intracranial magnetic resonance angiography (MRA), and postcontrast MRA encompassing the extracranial internal carotid artery (ICA) and vertebral artery (VA).

For enhancing images, patients were administrated Gd (Multihance, Bracco Diagnostics, Princeton, NJ, USA) at a dose of 0.2 mmol/kg, and postcontrast FLAIR images were obtained 5 min later. Postcontrast FLAIR images were obtained in the axial plane using a turbo spin-echo sequence with a section thickness of 5 mm, an intersection gap of 0.5 mm, a field of view (FOV) of 182 × 230 mm, repetition time (TR) /time to echo (TE) of 11,000/125 ms, and a matrix of 352 × 176.

### 2.3. Image Analysis

Two experienced neuroradiologists blinded to the clinical information of the patient performed the visual analysis of the MR images. An enhancing lesion was characterized by an enhancement on post-contrast T1WI. The enhancing lesions were subcategorized as subacute infarction according to other MR sequences and masses based on the DWI, ADC maps, and postcontrast FLAIR. The enhancement patterns on postcontrast FLAIR were subcategorized as follows. First, HARM was considered positive when cortical and sulcal contrast retention was observed. Second, peripheral rim enhancement was considered enhancement just outside the enhancing mass. Third, there was no enhancement on postcontrast FLAIR adjacent to the lesion.

### 2.4. Statistical Analysis

All findings were dichotomized (yes or no), and the prevalence of each finding was calculated for each observer. The interobserver agreement for each radiologic finding and HARM assessment was determined using Cohen’s kappa statistics. The strengths of agreement based on the κ values were categorized as follows: <0.20, poor; 0.21–0.40, fair; 0.41–0.60, moderate; 0.61–0.80, good; and 0.81–1.00, excellent. All statistical analyses were performed using SPSS software (version 24.0; SPSS Inc., Chicago, IL, USA).

## 3. Result

### 3.1. Patient Demographics

A total of 170 consecutive patients were enrolled in this study, including 94 patients in the subacute group and 76 patients in the mass group. The baseline characteristics of the study participants are summarized in Table 1 and Table 2. In the subacute infarction group, the mean age was 66.55 years and males accounted for 55.3%; in the mass group, the mean age was 57.23 years and males accounted for 44.7%. It took a median of 12.2 h (1–28 h) from the onset of neurologic events to the initial MR imaging evaluation. In the subacute infarction group, 54 patients (57.4%) had intracranial atherosclerotic lesions, and 20 patients (21.3%) had carotid steno-occlusive lesions. The location of the subacute infarction lesion was in the middle cerebral artery (MCA) territory in 68 patients (72.3%), the anterior cerebral artery (ACA) territory in 15 patients (16%), and the posterior cerebral artery (PCA) territory in 11 patients (11.7%). The mechanisms of stroke were as follows: 62 patients (66%) had a large artery-to-artery embolism; 10 (10.6%) had a cardiac embolism; 15 (16%) had a combined type; and 7 (7.4%) had an undetermined type.

### 3.2. Image Interpretation

For the initial imaging study, 83% (78/94) of the study participants presented with HARM in subacute cortical infarctions, and 63% (48/76) of the study participants presented with peripheral rim enhancement in subcortical masses. Among the patients with subcortical enhancing lesions, 51.8% (88/170) showed signs of HARM. Among the patients who were HARM-positive, 88.6% (78/88) were diagnosed with subacute infarction.

Among the 94 patients with subacute stroke, 48 (51%) had diffusion restrictions, and the signs of HARM on postcontrast FLAIR were found in 39 patients (81.2%) (Figure 1). Of the 46 patients (48.9%) without diffusion restriction, including only diffusion high signal intensity without ADC change, 39 (84.8%) showed signs of HARM on postcontrast FLAIR (Figure 2). Figure 3 shows a case in which it is difficult to distinguish between subacute infarction or a mass based on conventional MRI, including DWI, T2WI, and postcontrast T1WI. In other words, only the signs of HARM can be used to differentiate subacute infarction from the mass. Figure 4 shows a typical rim enhancement pattern around the cortical mass-like lesion.

The interobserver agreement was excellent for detecting the signs of HARM (κ = 0.88; 95% CI, 0.58–1.00) and peripheral rim enhancement (κ = 0.86; 95% CI, 0.56–1.00).

## 4. Discussion and Summary

In this article, we have attempted to improve the accuracy of the diagnosis of true subacute infarction and other mimics of subcortical enhancing lesions. For these reasons, we determined the feasibility of adopting HARM, which was originally considered a cerebral reperfusion marker for ischemic stroke. In this study, we demonstrated the clinical significance of HARM in differentiating subacute infarction from other subcortical enhancing masses. In other words, the presence of HARM was significantly associated with subacute infarction, and a peripheral rim enhancement pattern was observed around the mass on postcontrast FLAIR.

Traditionally, contrast extravasation has been termed HARM, which denotes the radiologic finding of a hyperintense signal within the cerebrospinal fluid (CSF) spaces visualized on postcontrast FLAIR images [34]. Classic HARM occurs at a mean time of 12.9 h after the onset of ischemia, and has been observed on the second MRI at 2–48 h after contrast administration in the first MRI in acute stroke patients [8]. In a previous study, “early” HARM (linear cortical enhancement near acute infarctions) was detected on postcontrast FLAIR images obtained 5 min after contrast administration, during the initial brain MRI for acute stroke, followed by widespread subarachnoid space enhancement. The time course of imaging findings of HARM suggests that the communicating channels of the pia mater were opened, allowing contrast material leakage from ischemic tissue into the subarachnoid space [17]. Regarding HARM in acute stroke, a majority of studies have focused on the risk of subsequent complications and clinical outcomes after thrombolysis or endovascular therapy. HARM is known to be associated with hemorrhagic transformation and poor clinical outcomes [31,34,35,36], although other studies have reported that there is no relation between HARM and hemorrhagic transformation [37,38]. Several studies found that patients with HARM may have worse neurologic recovery, and this may present the presence of HARM after thrombectomy as an indication of reperfusion injury. Multiple thrombectomy passes have been independently associated with a significant increase in BBB disruption, as evaluated by HARM [39,40,41,42]. HARM also facilitates the determination of ischemic injury even in the transient state without definite DWI lesions [43]. Recent studies have reported that 10% of patients with transient neurologic symptoms had HARM in their initial MRI study, which is helpful to confirm the ischemic insult [44]. HARM was found in 2.4% of patients with acute stroke or transient ischemic attack (TIA) without DWI lesions, and this result suggests HARM may help to prevent mis- or under-diagnosis in patients following a clinical course similar to TIA [45]. Furthermore, the presence of HARM was significantly associated with embolic infarctions, including large artery atherosclerosis and cardioembolic infarctions, which demonstrates that HARM can be useful for accurately classifying stroke etiology based on the TOAST classification [46].

MRI is more sensitive than CT in detecting ischemic changes [47,48], and different courses of MRI signal changes in different sequences are used as biomarkers of ischemic lesion age [49]. When experiencing an ischemic stroke, metabolic disturbances precipitate glial and neuronal transmembrane Na^+^/K^+^ transport failure, resulting in an immediate shift of Na^+^ and water from the extracellular to the intracellular space. An unchecked influx of Na^+^ and water results in osmotic expansion of the cell, thus, restricting the extracellular movement of water molecules. This is called cytotoxic edema, and represents the pathophysiology depicted by DWI minutes after a stroke [50,51,52]. In a previous report, there was a progressive increase in the net tissue water that could be depicted by T2 signal-based FLAIR imaging within the first few hours, which showed marked parenchymal hyperintensity [53]. During the acute stage, there is further worsening of cytotoxic edema and increased tissue water, resulting in the prolongation of T1 and T2 relaxation, the effacement of convexity sulci, and mild swelling of the gyri without a mass effect. During the subacute stage, there is an increase in extracellular fluid (vasogenic edema) because of a breakdown in the BBB and the rupture of swollen cells. During this phase, imaging shows increased edema, a mass effect, and possible herniation, depending on the size and the site of the infarct. The incidence of hemorrhagic transformation caused by a combination of vascular injury, reperfusion, and altered permeability is the highest during the subacute stage [54]. However, these typical imaging findings are not always seen, and MRI shows only nonspecific findings or false-negative findings in some cases, therefore, clinical decision making remains challenging.

Sensitivity in detecting ischemia may be increased through the addition of paramagnetic contrast agents, most likely due to either accentuation of the underlying flow derangement or the accumulation of abnormal amounts of contrast agent in the ischemic tissue [1]. In a prospective series of MRI during the first week after the onset of stroke, intravascular enhancement was a relatively common finding on the first and second days, and parenchymal enhancement in the cortical, subcortical, and deep parenchymal structures was more frequent at later time points than during the acute phase [6]. Parenchymal contrast enhancement, most likely the result of an immature BBB caused by neovascularity during the subacute phase of complete ischemia, appears as late as several days after the onset of stroke, and is maximal at the end of the first week. The delay has been explained as being due to the time required for angiogenesis in the granulation tissue [2,6,54]. In some cases, intense parenchymal enhancement may resemble enhancement in tumors, making diagnosis difficult in patients with an obscure clinical history. In these situations, DWI may be useful in distinguishing between a tumor and an infarct, however, it may fail to detect subacute infarcts because pseudo-normalization of the ADC can mask the lesion on these images [6,55,56]. Since DWI is not necessarily conclusive, it is important to be aware of various patterns and the temporal evolution of contrast enhancement in distinguishing an infarct from a non-ischemic process. In confusing cases, short-term follow-up examinations to look for progression or reduction in lesion enhancement can help clinicians make an accurate diagnosis [6]. We demonstrated that HARM will be helpful in ambiguous cases.

Contrast agents are frequently used to improve the lesion characterization of CNS pathologies by utilizing the T1-shortening effect. Gd is the most commonly used agent in CNS disorders, and has a shortening effect on both T1 and T2 relaxation times of tissue at clinical doses [57,58]. Contrast enhancement in the CNS is the result of a combination of three processes. For intra-axial lesions, the BBB should be disrupted for Gd to enter the extracellular space. For extra-axial lesions, enhancement is observed in lesions with relatively high vascularity. Furthermore, for leptomeningeal lesions, there is contrast leakage from the vessels into the CSF [3,4,8,18]. Based on these processes, T1WI is typically used for contrast examinations. Various pathological conditions result in abnormal contrast enhancement in the brain. Neoangiogenesis, active inflammation, cerebral ischemia, and pressure overload disorders, such as eclampsia and hypertension, are all associated with alterations in BBB permeability [18].

There are different mechanisms of enhancement for postcontrast T1WI and FLAIR images, which can be explained by a combination of a different T1-shortening effect depending on the amount of Gd, and a different T2 effect based on the vascularity of the disorders [5,15]. A phantom study by Lee et al. [8] showed that FLAIR was more sensitive to T1-shortening than T1WI at lower concentrations of Gd (0.2 mmol). In addition, normal vascular structures and meninges cannot be visualized on postcontrast FLAIR, unlike postcontrast T1WI. This evidence indicates that faint enhancement on postcontrast T1WI may be depicted more clearly on postcontrast FLAIR at a clinical dose.

Postcontrast FLAIR demonstrates a typical peripheral rim enhancement pattern related to the vascular supply of the tumor, which is more commonly observed in larger masses (>2 cm in diameter). For the evaluation of masses, the highly vascular central portion of the mass is strongly enhanced on postcontrast T1WI, whereas a high concentration of Gd in the central portion induces signal loss on postcontrast FLAIR images. The less vascular capsule, supplied by pial arteries, may have a lower concentration of Gd, resulting in a peripheral rim enhancement pattern on postcontrast FLAIR. However, in tumors less than 0.5 cm in diameter from our research, this effect is masked, and only homogenous enhancement is shown. These results are more sensitive because previous studies have demonstrated that tumors less than 2 cm in diameter are the standard for the visualization of peripheral rim enhancement patterns [59].

This study had several limitations. First, this study was a retrospective single-center study involving a small cohort, and it had a risk of selection bias. Although it enrolled consecutive participants, prospectively confirming the research findings would provide more concrete evidence. Second, we did not analyze the severity of HARM to discern if there is a gradation of injury that may further correlate with the final diagnosis. Third, there is a limitation in not being able to match gender and age between infarction and mass groups.

Despite these limitations, we have shown that contrast enhancement patterns may differ with the cortical lesion characteristics. In other words, linear sulcal enhancement, as observed with the HARM, is highly associated with subacute infarction, and rim enhancement around the cortical lesion is highly associated with mass lesions. From these results, when clinicians find it difficult to diagnose cortical enhancing lesions, the enhancement patterns on postcontrast FLAIR can facilitate accurate diagnosis and guide treatment. In conclusion, we found that HARM with enhancing subcortical nodules was highly suggestive of subacute infarctions, and recommended a postcontrast FLAIR sequence for differentiating the ischemic subcortical infarctions from malignant lesions such as small metastasis.

## Figures and Tables

**Figure 1 diagnostics-11-02161-f001:**
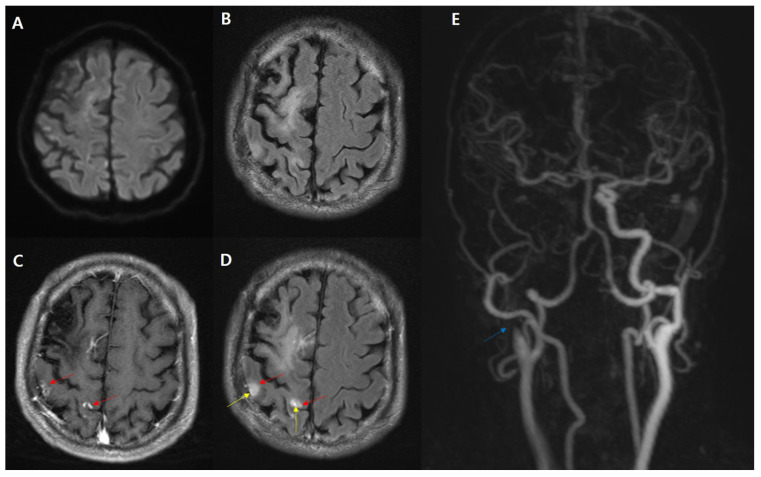
Typical HARM related to subacute infarction. A 65-year-old male presented with back pain and left side weakness after slipping in his house. DWI (**A**) with low signal intensity on the ADC map (not shown) showing a barely discernible hyperintense sign. Postcontrast T1WI (**C**) showing multiple nodular enhancement at the right parietal cortex (red arrow). A postcontrast FLAIR image (**D**) showing a linear enhancement along the adjacent right parietal sulci (yellow arrow), which is well-visualized as compared with the non-enhanced FLAIR (**B**) consistent with the HARM. Carotid TOF-MRA showing occlusion of the right proximal ICA (blue arrow, (**E**)).

**Figure 2 diagnostics-11-02161-f002:**
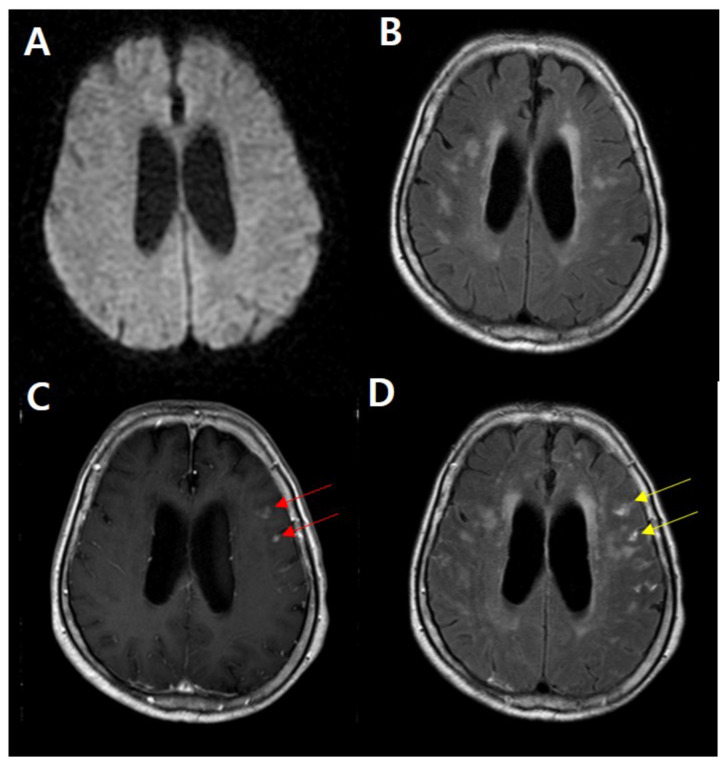
HARM related to subacute infarction. A 67-year-old male presented with a transient language problem and right arm weakness with preceding thunderclap headache, approximated 10 days. Initial DWI (**A**) showing no diffusion restriction lesion in the brain. Postcontrast T1WI (**C**) showing two nodular enhancements at the left frontal cortex (red arrow). Postcontrast FLAIR image (**D**) showing the typical HARM (yellow arrow), which is well-visualized compared with the non-enhanced FLAIR (**B**).

**Figure 3 diagnostics-11-02161-f003:**
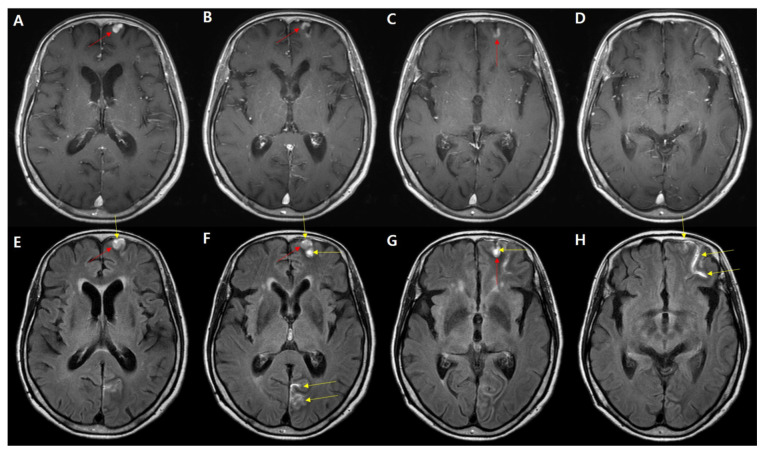
HARM related to a cortical enhancing lesion. A 76-year-old female presented with gait disturbance for 2 weeks. (**A**–**D**) Postcontrast T1WI showing nodular enhancement at the left anterior frontal cortex (red arrow). A postcontrast FLAIR image showing enhancement at the left frontal cortex and another linear, and stronger, enhancement along the left frontal sulci and left occipital sulci (yellow arrow, (**E**–**H**)). Based on the HARM, this patient was treated with an antiplatelet and aspirin for 3 months, and her symptoms improved.

**Figure 4 diagnostics-11-02161-f004:**
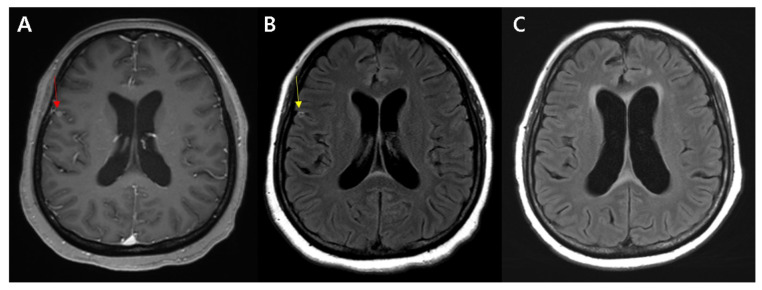
Rim enhancement superficial to the metastatic brain tumors. A 55-year-old female presented with headache showed focal homogeneously enhancing mass at right precentral gyrus, unknown cause (red arrow, (**A**)). Peripheral rim enhancement around the mass is observed on postcontrast FLAIR (yellow arrow, (**B**)), which means the cortical lesion is a mass rather than an infarction. The patient was diagnosed with lung cancer through further examination, including bronchoscopic lung biopsy, and we concluded that the enhancing lesions were metastasis from lung cancer. Subsequently, she received whole brain radiotherapy, and was started on treatment with combination chemotherapy. Follow up MRI after 3 months shows complete tumor disappearance (**C**).

**Table 1 diagnostics-11-02161-t001:** Clinical and radiologic characteristics of subacute infarction.

	Subacute Infarction
Average age [y]	66.55 ± 11.3
Gender	
Male	52 (55.3)
Female	42 (44.7)
Hypertension	69 (73.4)
Diabetes mellitus	43(45.7)
Hyperlipidemia	48 (51.0)
History of CAD	16 (17.0)
Current smoker	30 (31.9)
Alcohol	20 (21.3)
Previous antiplatelet	
none	72 (76.6)
SAPT	13 (13.8)
DAPT	8 (8.5)
Previous statin	15 (16.0)
BMI (kg/m^2^)	24.28 ± 4.3
SBP (mmHg)	148 ± 18.4
DBP (mmHg)	88 ± 10.2
Initial NIHSS	4.22
Infarction territory	
MCA	68 (72.3)
ACA	15 (16.0)
PCA	11 (11.7)
Vessel conditions	
Intracranial occlusion	12 (12.8)
Intracranial stenosis	42 (44.7)
Carotid occlusion	10 (10.6)
Carotid stenosis	10 (10.6)
No stenoocclusive	20 (21.3)
The mechanism of stroke	
Large artery atherosclerosis	62 (66.0)
Cardiac embolism	10 (10.6)
Combined type	15 (16.0)
Undetermined type	7 (7.4)

Results presented as number and percent (% column). CAD: coronary artery disease, SAPT: single antiplatelet treatment, DAPT: dual antiplatelet treatment, BMI: body mass index, SBP: systolic blood pressure, DBP: diastolic blood pressure, NIHSS: National Institutes of Health Stroke Scale, MCA: middle cerebral artery, ACA: anterior cerebral artery, PCA: posterior cerebral artery.

**Table 2 diagnostics-11-02161-t002:** Clinical characteristic of mass dataset.

	Mass
Average age [y]	57.23 ± 12.1
Gender	
Male	34 (44.7)
Female	42 (55.3)
No. of Metastasis	66 (86.9)
Origin of Metastasis	
Lung	42
Breast	19
Melanoma	2
Genitourinary	1
Gastrointestinal	1
Miscellaneous	1
No. of Malignant glioma	7 (9.2)
No. of Lymphoma	3 (3.9)

Results presented as number and percent (% column).

## Data Availability

The data that support the findings of current study are available from the corresponding author on reasonable request.

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
