# Peer review of "The Clinical Significance of the Hyperintense Acute Reperfusion Marker Sign in Subacute Infarction Patients"

_diagnostics, 2021, doi:10.3390/diagnostics11112161_

Round 1

Reviewer 1 Report

  1. Lines 72-73: This is an extremely common practice for obtaining DWI and ADC with all MRI sequences. PWI is not routinely obtained.
  2. Lines 73-75: MRI DWI with ADC is highly sensitive in picking up acute ischemic strokes with 1.5 and 3 T MRIs. These above-mentioned lines are somewhat misleading.
  3. Lines 53 – 63: Post-contrast FLAIR is not a common practice. Besides, subacute strokes on MRI within 14 days still have DWI changes with variable ADC correlate. Post-contrast enhancement pattern is largely different between subacute stroke and other malignant lesions and has been extensively studied, not necessarily on postcontrast FLAIR sequences.
  4. Line 92: How did the authors exclude acute infarction patients? Clinical or radiological findings?
  5. Did the authors exclude patients with a known diagnosis of brain tumors or Mets, who underwent ‘stroke protocol’? Or patients with a known diagnosis of demyelinating conditions? Seems like the authors did not exclude such patients first, but excluded some patients afterward based on imaging. This is a very strong source of selection bias in the study.
  6. Lines 100-102: unclear. Did authors obtain MRI more than once?
  7. Lines 98-99: Excluded patients with subacute infarction in most locations except subcortical. So, the results should be generalized to all subacute strokes! Results are leading.
  8. Line 158: final number of patients in the study analysis is different than what was previously mentioned in the study.
  9. Figure 5: This is a violation of your inclusion criteria. The authors wanted to include patients with infarction and seem like there are certain patients who had a hemorrhage. Hemorrhage post-contrast pattern certainly different than others. Such patients with hemorrhage should have been excluded from the study.
  10. HARM has been studied before in primary hemorrhage; the concept is not very popular though. But this contradicts the authors’ study. article - 10.1212/WNL.0b013e318236ef46
  11. The article overestimates the HARM importance in subacute subcortical strokes, which is a very selective population stroke population. The results cannot be applied to all subacute strokes.

Author Response

Dear Editors and Reviewers

We are very grateful for the insightful comments from the reviewers. I and my corresponding author are very happy to have a chance to re-submit our manuscript. We did our best to give the valid and convincing explanations to the reviewers in this letter, and revised the manuscript following the reviewers’ comments.

The followings are the point-by-point response to the comments.

Reviewer A's comments #1 &2> Lines 72-73: This is an extremely common practice for obtaining DWI and ADC with all MRI sequences. PWI is not routinely obtained.

Lines 73-75: MRI DWI with ADC is highly sensitive in picking up acute ischemic strokes with 1.5 and 3 T MRIs. These above-mentioned lines are somewhat misleading.

Reply1&2: Thank you for your comments. According to your suggestion, we’ve modified the paragraphs as follows:

Changes in the text: Although DWI has been shown to be an excellent tool for the detection of acute stroke, its sensitivity would appear to have limitations, as witnessed by reports of false-negative cases. The prevalence rate of false negative DWI in patients with acute ischemic stroke is highly variables, ranging from 0% [19] to 25.6% [20-29], notwithstanding that the stroke lesions in these studies were not always confirmed radiologically. Furthermore, ischemic stroke mimics can occur on DWI with non-cerebrovascular neurological conditions [30] and this can make diagnosis even more challenging. Moreover, PWI can also help prevent misinterpretation, this study is not routinely performed without clinical suspicion.

Reviewer A's comments #3> Lines 53 – 63: Post-contrast FLAIR is not a common practice. Besides, subacute strokes on MRI within 14 days still have DWI changes with variable ADC correlate. Post-contrast enhancement pattern is largely different between subacute stroke and other malignant lesions and has been extensively studied, not necessarily on postcontrast FLAIR sequences.

Reply: Thank you very much for your comments. We sympathize with the comments. As you said, most stroke cases are accurately diagnosed with DWI, ADC maps, and postcontrast T1WI. In this study, we paid attention to the diagnosis of less prevalent stoke cases of tricky diagnostic problems with a pattern of nodular enhancing lesions in subcortical and cortical parenchyma which had similar features of malignant masses such as a small metastatic lesion. We think that the conventional process has a risk of misdiagnosis in stroke if enhanced FLAIR had not been used. Our findings suggested that postcontrast FLAIR sequences may have the chance to improve correct diagnosis of less prevalent and challenging stroke cases.

Reviewer A's comments #4> Line 92: How did the authors exclude acute infarction patients? Clinical or radiological findings?

Reply: Thank you for your comments. We agree that the sentence was not detailed enough. In this study, acute stroke was defined based on radiologic and clinical findings. We classified patients with diffusion restriction on ADC maps and DWI (low signal intensity on ADC maps and high signal intensity on DWI), with no parenchymal enhancement on postcontrast T1WI, and with neurologic symptom within 7 days of onset as acute stroke patients and excluded from the study. To clarify this, we have added the details of definition and exclusion criteria regarding acute infarction as follows:

Changes in the text: Strokes may be classified and dated as early hyperacute, late hyperacute, acute, subacute, or chronic. As a first step, we needed to agree on a common framework for defining what is meant by “acute” and “subacute” stroke. There is substantial heterogeneity in the terminology denoting time from onset of stroke. The definition of hyper-acute as 0-24 hours, acute as 1-7 days was affirmed by the international Stroke Recovery and Rehabilitation Roundtable, which link to the currently known biology of recovery [20, 21]. ADC maps may depict darkening within minutes of stroke onset and are more sensitive than diffusion weighted sequences. A good rule of thumb is that if signal intensity on ADC maps is low, the stroke is less than 1 week old. Low signal intensity on ADC maps persists for about 7-10 days and it can be observed in early subacute stage. In most infarcts, parenchymal enhancement on postcontrast T1WI is seen between 1 week and 2 months after stroke [22]. In this study, acute stroke was defined based on these radiologic findings and clinical symptoms. We classified the patients with diffusion restriction (low signal intensity on ADC maps and high signal intensity on DWI), with no parenchymal enhancement on postcontrast T1WI, and with neurologic symptom within 7 days of onset as having acute stroke and excluded those cases from the study.

Reviewer A's comments #5> Did the authors exclude patients with a known diagnosis of brain tumors or Mets, who underwent ‘stroke protocol’? Or patients with a known diagnosis of demyelinating conditions? Seems like the authors did not exclude such patients first, but excluded some patients afterward based on imaging. This is a very strong source of selection bias in the study.

Reply: Thank you for your comments. We intended to compare subacute infarction group with mass group in this study, and selected those two groups separately. For the selection of subacute infarction group, patients underwent our “Acute stroke protocol” were identified first. Among them, we excluded cases with brain tumors or metastatic lesions or demyelinating disease (ex. subacute infarction + demyelinating disease) based upon additional imaging findings (Figure 1). For the selection of mass group, we separately selected the patients who underwent ‘brain tumor protocol’ not ‘stroke protocol’, and showed cortical/subcortical enhancing masses during the same period.

We have made changes to clarify this.

Changes in the text: During the same period, a total of 76 patients diagnosed with enhancing masses who underwent our “Brain tumor protocol” were also identified. The masses located in the cortex or subcortical white matter were analyzed. The enhancing masses were subcategorized as metastasis (n=66), malignant glioma (n=7), and lymphoma (n=3).

Reviewer A's comments #6> Lines 100-102: unclear. Did authors obtain MRI more than once?

Reply: Thank you for your comment. We agree that the sentence is unclear. We characterized pathology based on the typical MRI findings, clinical courses and stroke treatment outcomes for a minimum of 1 month. Follow-up MRI were obtained more than once if necessary in challenging cases. We have made some changes as follows:

Changes in the text: We characterized pathology based on the typical MRI findings, clinical courses and stroke treatment outcomes for a minimum of 1 month. Follow-up MRI were obtained more than once if necessary in challenging cases.

Reviewer A's comments #7> Lines 98-99: Excluded patients with subacute infarction in most locations except subcortical. So, the results should be generalized to all subacute strokes! Results are leading.

Reply: Thank you very much for your comment. The main concern of this study is the differential diagnosis of nodular enhancing lesions in the cortical and subcortical areas, because it is more challenging than cases with lesions in other locations. A pattern of nodular enhancing lesions in subcortical and cortical parenchyma is typical for hematogenous dissemination of metastatic neoplasms and clot emboli. Small cortical/subcortical infarctions with nodular enhancement share morphologic features with these disease entities and clinicians have difficulty making clinical decisions. We insisted that HARM can be used as a clue in diagnosis of infarction when the clinicians facing these kind of challenging cases. Therefore, we limited the study population to the patients with cortical/subcortical subacute infarction.

Reviewer A's comments #8> Line 158: final number of patients in the study analysis is different than what was previously mentioned in the study.

Reply: A total of 170 consecutive patients were enrolled in this study, including 94 patients in the subacute group and 76 patients in the mass group. We agreed that the sentence was ambiguous. So we modified the sentence as follows:

Changes in the text: A total of 170 consecutive patients were enrolled in this study, including 94 patients in the subacute group and 76 patients in the mass group.

Reviewer A's comments #9> Figure 5: This is a violation of your inclusion criteria. The authors wanted to include patients with infarction and seem like there are certain patients who had a hemorrhage. Hemorrhage post-contrast pattern certainly different than others. Such patients with hemorrhage should have been excluded from the study.

Reviewer A's comments #10> HARM has been studied before in primary hemorrhage; the concept is not very popular though. But this contradicts the authors’ study. article - 10.1212/WNL.0b013e318236ef46

Reply #9&10: Thank you very much for your comment. We agree and have made a change in figure 5. Now a case of metastatic mass with typical peripheral rim enhancement and with no HARM on postcontrast FLAIR is illustrated. We think that this case represents the mass group better.

Reviewer A's comments #11> The article overestimates the HARM importance in subacute subcortical strokes, which is a very selective population stroke population. The results cannot be applied to all subacute strokes.

Reply: Thank you very much for your comment. We agree that our results cannot be applied to all subacute stroke patients. However, our results suggest that HARM can be a useful marker for accurate diagnosis of stroke in selected patients. We think that HARM still has a diagnostic value in that it prevents inappropriate treatment even in small percentage of patients.

Reviewer 2 Report

This manuscript reported that the presence of HARM can be used to differentiate subacute infarctions from intracranial masses. It is interesting and useful to alert clinicians to use postcontrast FLAIR to tell one from the other. There are a few issues to clarify:

  1. The major issue here is the differentiation between subacute infarction and masses. However, the Results section described findings from subacute infarction and masses, but the images presented findings from intracranial haemorrhage which is not even mentioned in the demographics of patients or in Table 2. This should be consistently presented.
  2. Age seems to be a confusing variable. The mean age of each group is significantly higher than that of the males or females. The difference in age between males and females might be confounding the results. Please touch on this and may include in the limitations part.
  3. The informed consent was inconsistently stated in the Materials section and the Acknowledgement section. One is waived and the other obtained. 
  4. The selection of patients with <0.5cm or <2cm nodular lesions led to the conclusion of usefulness of postcontrast FLAIR in discriminating subacute infarction from masses, how about larger lesions? Are there any literatures supporting the usefulness of postcontrast FLAIR in discriminating subacute infarction from masses? or maybe other lesions? Discussion on this might be more appropriate for non-expert readers to understand.
  5. How do you define acute or subacute infarction, references are necessary to add. 0-6 hours or 24 hours or 7 days as acute stroke? 
  6. There is one error showing the occlusion in the ICA in figure 2, which should be placed in figure 3.
  7. Figure 5D does not have a label in the legend though I guess it denotes the resolution of haemorrhage.

Author Response

Reviewer B's comments #1> This manuscript reported that the presence of HARM can be used to differentiate subacute infarctions from intracranial masses. It is interesting and useful to alert clinicians to use postcontrast FLAIR to tell one from the other. There are a few issues to clarify: The major issue here is the differentiation between subacute infarction and masses. However, the Results section described findings from subacute infarction and masses, but the images presented findings from intracranial hemorrhage which is not even mentioned in the demographics of patients or in Table 2. This should be consistently presented.

Reply: Thank you very much for your comment. We agree and have made a change in figure 5. Now a case of metastatic mass with typical peripheral rim enhancement and with no HARM on postcontrast FLAIR is illustrated. We think that this case represents the mass group better.

Reviewer B's comments #2> Age seems to be a confusing variable. The mean age of each group is significantly higher than that of the males or females. The difference in age between males and females might be confounding the results. Please touch on this and may include in the limitations part.

Reply: According your comments, we have added the sentence in the limitation part.

Changes in the text: There is a limitation in not being able to match gender and age between infarction and mass groups.

Reviewer B's comments #3> The informed consent was inconsistently stated in the Materials section and the Acknowledgement section. One is waived and the other obtained.

Reply: Thank you very much for your comments. We have changed the contents of Materials section as follows:

Changes in the text: This retrospective, single-center study was approved by the institutional review board, and the requirement for informed consent was waived. informed consent was obtained from all subjects involved in the study.

Reviewer B's comments #4> The selection of patients with <0.5cm or <2cm nodular lesions led to the conclusion of usefulness of postcontrast FLAIR in discriminating subacute infarction from masses, how about larger lesions? Are there any literatures supporting the usefulness of postcontrast FLAIR in discriminating subacute infarction from masses? or maybe other lesions? Discussion on this might be more appropriate for non-expert readers to understand.

Reply: In the case of a large mass, the difference in enhancement pattern is clear in the first place, so there is less need to differentiate it by postcontrast FLAIR. Therefore, in this study, we focused on the small-sized lesions.

Reviewer B's comments #5> How do you define acute or subacute infarction, references are necessary to add. 0-6 hours or 24 hours or 7 days as acute stroke? 

Reply: Thank you very much for your comment. According to your opinion, we have added the definition of terminology denoting time from onset of stroke and references as follows:

Changes in the text: Strokes may be classified and dated as early hyperacute, late hyperacute, acute, subacute, or chronic. As a first step, we needed to agree on a common framework for defining what is meant by “acute” and “subacute” stroke. There is substantial heterogeneity in the terminology denoting time from onset of stroke. The definition of hyper-acute as 0-24 hours, acute as 1-7 days was affirmed by the international Stroke Recovery and Rehabilitation Roundtable, which link to the currently known biology of recovery [20, 21]. ADC maps may depict darkening within minutes of stroke onset and are more sensitive than diffusion weighted sequences. A good rule of thumb is that if signal intensity on ADC maps is low, the stroke is less than 1 week old. Low signal intensity on ADC maps persists for about 7-10 days and it can be observed in early subacute stage. In most infarcts, parenchymal enhancement on postcontrast T1WI is seen between 1 week and 2 months after stroke [22]. In this study, acute stroke was defined based on these radiologic findings and clinical symptoms. We classified the patients with diffusion restriction on ADC maps and DWI (low signal intensity on ADC maps and high signal intensity on DWI), with no parenchymal enhancement on postcontrast T1WI, and with neurologic symptom within 7 days of onset as having acute stroke and excluded those cases from the study.

Reviewer B's comments #6> There is one error showing the occlusion in the ICA in figure 2, which should be placed in figure 3.

Reply: Thank you very much for your comment. We found that position of Figure 2 and Figure 3 changed. We have corrected the order of Figure 2 and Figure 3.

Reviewer B's comments #7> Figure 5D does not have a label in the legend though I guess it denotes the resolution of haemorrhage.

Reply: Thank you for your comment. We have changed the case of Figure 5 to another case.

Thank you very much for your comments.

Round 2

Reviewer 1 Report

Dear Authors,

Thank you for your detailed explanation of my comments. I have carefully reviewed all your comments. 

This study has a really good concept however, it has a lot of inconsistency in patient selection (more than one source of selection bias), implementation of the consistent intervention (MRI which has been obtained more than once on some patients),  and improper generalization and overestimation of the results.  I am not convinced with the study design. The authors should thoroughly decide their target population. I would recommend a major revision with consistent study inclusion-exclusion criteria to eliminate selection bias, fixed intervention across the study, and precise results. 

Author Response

Dear  Reviewer

I and my corresponding author thought about your comment again. At the beginning of the study design, we were very worried about not being able to come to a conclusion for figure 4 case. So, we analyzed retrospectively on the differentiation of cortical enhancing lesions, and we could conclude that post-contrast FLAIR sequence taken at our hospital is helpful.

The description of the patient selection process was written in a slightly confusing way, contrary to the intention, so it has been modified as follows and delete the figure 1 flow chart.

Change in the manuscript: 

Between May 2019 and May 2021, we retrospectively reviewed a total of 12,582 MRIs for searching the cortical enhancing lesions. Based on the picture archiving communication system (PACS) and electronic medical records (EMR) results, 275 patients who had enhancing lesions in the brain were identified and were subdividied into the stroke (199 patients) and brain tumors (76 patients). Among 199 patients who had diagnosed infarction, a total of 69 patients were excluded for the following reasons: 1) 26 had been diagnosed with acute infarction, 2) 6 patients had an enhancing mass combined with infarction, 3) 1 patient had meningitis, 4) 1 patient had demyelinating disease, 5) 3 patients had a previous history of surgery, and 6) the brain lesion in 32 patients did not meet the available reference standard due to poor image quality with motion artifact. Furthermore, 36 patients were additionally excluded because subacute infarctions were located in the periventricular white matter, deep gray matter, or infratentorium, including the brainstem and cerebellum. Subsequently, 94 patients with subacute cortical infarction were included. We characterized pathology based on the typical MRI findings, clinical courses and stroke treatment outcomes for a minimum of 1 month. Follow-up MRI were obtained more than once if necessary in challenging cases. The details of this cohort are shown in Table 1. During the same period, a total of 76 patients diagnosed with enhancing masses were also identified. The masses located in the cortex or subcortical white matter were analyzed. The enhancing masses were subcategorized as metastasis (n=66), malignant glioma (n=7), and lymphoma (n=3).

We did our best to give the valid and convincing explanations to the reviewers in this letter.

Based on the reviewer's comments, future prospective research is in progress, and I can write a very good thesis and the revierwer's comments were very helpful for us. Thank you very much again.

Round 3

Reviewer 1 Report

The authors have provided valid and reasonable explanations to all my comments. The authors have made major changes in the manuscript. The article looks much better. I do not have any further high level comments. Thank you for your hard work. 

Author Response

thank you